# DictPFL: Efficient and Private Federated Learning on Encrypted Gradients

**Jiaqi Xue[1], Mayank Kumar[1], Yuzhang Shang[1], Shangqian Gao[2], Rui Ning[3]**
**Mengxin Zheng[1], Xiaoqian Jiang[4], Qian Lou[1]**
[1]University of Central Florida    [2]Florida State University    [3]Old Dominion University
[4]University of Texas Health Science Center at Houston
`{jiaqi.xue,mayank.kumar,yuzhang.shang,mengxin.zheng,qian.lou}@ucf.edu`
`sgao@cs.fsu.edu, rning@odu.edu, xiaoqian.jiang@uth.tmc.edu`

## Abstract

Federated Learning (FL) enables collaborative model training across institutions without sharing raw data. However, gradient sharing still risks privacy leakage, such as gradient inversion attacks. Homomorphic Encryption (HE) can secure aggregation but often incurs prohibitive computational and communication overhead. Existing HE-based FL methods sit at two extremes: encrypting all gradients for full privacy at high cost, or partially encrypting gradients to save resources while exposing vulnerabilities. We present **DictPFL**, a practical framework that achieves full gradient protection with minimal overhead. DictPFL encrypts every transmitted gradient while keeping non-transmitted parameters local, preserving privacy without heavy computation. It introduces two key modules: **Decompose-for-Partial-Encrypt (DePE)**, which decomposes model weights into a static dictionary and an updatable lookup table—only the latter is encrypted and aggregated, while the static dictionary remains local and requires neither sharing nor encryption; and **Prune-for-Minimum-Encrypt (PrME)**, which applies encryption-aware pruning to minimize encrypted parameters via consistent, history-guided masks. Experiments show that DictPFL reduces communication cost by **402–748×** and accelerates training by **28–65×** compared to fully encrypted FL, while outperforming state-of-the-art selective encryption methods by **51–155×** in overhead and **4–19×** in speed. Remarkably, DictPFL's runtime is within **2×** of plaintext FL, demonstrating—for the first time—that HE-based private federated learning is practical for real-world deployment. The code is publicly available at `https://github.com/UCF-ML-Research/DictPFL`.

## 1 Introduction

Federated Learning (FL) [1] was introduced to enable collaborative training of a shared machine learning model among different data owners (e.g., hospitals or banks), where model gradients (or weights), rather than raw data, are shared to address privacy concerns. However, even sharing gradients poses privacy risks, as attackers could potentially exploit this information. For instance, model inversion (or gradient inversion) attacks [2, 3] have demonstrated the feasibility of reconstructing a client's original training data from the gradients shared by clients. In such scenarios, the server or users with access to the server can act as potential attackers.

To protect the privacy of clients' gradients during aggregation and enable private FL, various privacy-preserving primitives such as Differential Privacy (DP) [4, 5], Secure Multiparty Computation (MPC) [6, 7], and Homomorphic Encryption (HE) [8–20] have been utilized. Among these methods, HE is especially appealing in cross-silo settings [21], as it provides non-interactive privacy

protection without the accuracy-privacy trade-off associated with DP [22–24] and without requiring the assumption of trusted servers, as in MPC [25–28]. In HE-based privacy-preserving federated learning [21, 29–31], locally updated gradients are encrypted by clients before being shared with the server, allowing the server to perform homomorphic aggregation directly on ciphertexts. Despite its security benefits, HE introduces significant overhead: ciphertext expansion increases communication costs by 1 to 3 orders of magnitude, while encryption, decryption, and homomorphic aggregation impose high computational costs [21, 31].

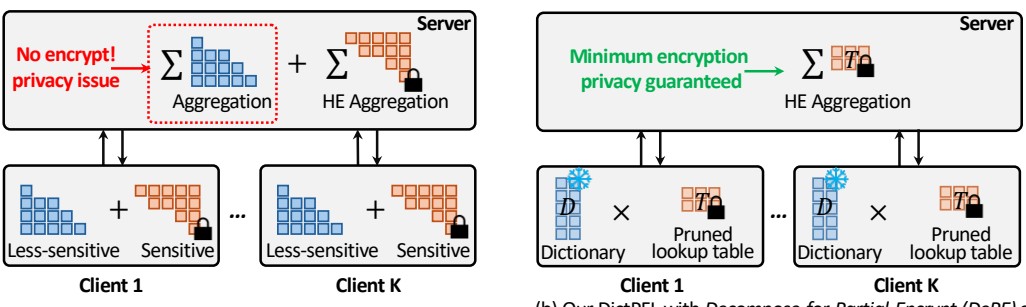

(a) Prior HE-based PPFL with *Select-and-Encrypt (SaE) strategy*

(b) Our DictPFL with *Decompose-for-Partial-Encrypt (DePE) and Prune-for-Minimum-Encrypt (PrME) strategy*

Figure 1: (a) Prior HE-based FL [31] encrypts only deemed sensitive gradients. The less-sensitive weights are shared in plaintext, which may lead to privacy concerns. (b) In contrast, our DictPFL minimizes encryption while ensuring privacy guarantees through Decompose-for-Partial-Encrypt (DePE) and Prune-for-Minimum-Encrypt (PrME). DePE decomposes gradients into a frozen dictionary and a trainable lookup table, with only the lookup table being encrypted and shared for aggregation. PrME further prunes the lookup table parameters on the client side to reduce encryption costs.

Prior efforts to improve the efficiency of HE-based FL often compromise privacy. The state-of-the-art method, FedML-HE [31], as illustrated in Figure 1 (a), adopts a *Select-and-Encrypt (SaE)* strategy: clients pre-compute sensitivity scores for model parameters and encrypt only the gradients of the most sensitive subset (e.g., the top 10%), while transmitting the remaining parameters in plaintext. However, these unencrypted gradients still expose private information. As shown in Figure 7 (a), when 30% of gradients remain unencrypted under FedML-HE, gradient inversion attacks can reconstruct images with up to 23% similarity to the originals. Moreover, the pre-computed sensitivity scores fail to capture dynamic sensitivity shifts during training, as parameter updates continually alter their privacy relevance. Consequently, encrypting all transmitted gradients remains essential to eliminate leakage. Although the SaE strategy achieves lower communication overhead and faster training than fully encrypted methods, it inevitably exposes privacy risks due to the shared plaintext gradients.

To address this challenge, we propose DictPFL, as shown in Figure 1 (b), which ensures that all shared parameters are fully encrypted to guarantee privacy while minimizing the number of shared parameters through two modules: *Decompose-for-Partial-Encrypt (DePE)* and *Prune-for-Minimum-Encrypt (PrME)*. *DePE* decomposes the model weights to be trained into a globally consistent dictionary, which is identical across all clients, and a lookup table, which each client trains independently. Only the encrypted gradients of the lookup table are shared with the server for aggregation, while the globally consistent dictionary remains frozen and is *never* shared. Building on *DePE*, *PrME* further reduces encrypted lookup tables through consistent pruning across clients. Unlike plaintext-level pruning in FL [32–34], where clients perform pruning locally and the server aligns the retained gradients before aggregation, HE-based FL presents unique challenges: retained gradients are batch-encrypted into ciphertexts in a SIMD format, preventing the server from aligning them without decryption. *PrME* addresses this by pruning based on shared global gradient history, ensuring consistent indices. Additionally, dynamic probabilities are assigned to the pruned parameters, allowing for their potential reintroduction in future rounds and mitigating the negative effects of premature pruning. Since the pruned lookup tables are significantly smaller than the full model weights, and all transmissions are encrypted, this approach substantially reduces the number of ciphertexts without compromising privacy.

Extensive experiments demonstrate that **DictPFL** delivers substantial performance gains over the state-of-the-art FedML-HE [31] across diverse tasks, including (i) image recognition, (ii) text classification,

and (iii) text generation. Compared with fully encrypted frameworks [35], DictPFL outperforms the selectively encrypted FedML-HE [31] by lowering communication overhead by **51–155×** and speeding up training by **4–19×**. Remarkably, DictPFL introduces less than a **2×** training-time increase even relative to plaintext FL, demonstrating that homomorphic encryption—commonly considered prohibitively expensive—can in fact be practical for federated learning at scale.

## 2 Background and Motivation

### 2.1 Privacy-preserving Federated Learning

Federated Learning (FL) enables collaborative training among distributed clients without directly sharing their datasets. In this framework, clients train their models locally and send gradients (or model updates) to a central server, which aggregates them using algorithms such as FedAvg [36] and FedSGD [1]. However, the direct exposure of local gradients to the server poses severe privacy risks [37]. For instance, with access to a client's local gradients, the server can perform model inversion attacks [2, 3, 38] to reconstruct the client's dataset.

Several methods have been proposed to protect the gradients transmitted between clients and the server. One strategy employs Differential Privacy (DP) [22–24] by injecting noise into the gradients before sharing them. Although DP imposes minimal computational overhead, it inevitably degrades model performance due to the added noise. Secure Multi-Party Computation (MPC) [25, 28] ensures that the server can access only aggregated gradients rather than individual ones. However, the aggregated gradients remain exposed to the server, and the reliance on multiple non-colluding servers makes MPC unsuitable for single-server settings.

Another approach leverages Homomorphic Encryption (HE) [21, 31] to encrypt gradients on the client side, enabling the server to aggregate them without decryption. HE provides end-to-end protection by securing gradient transmission, aggregation, and server storage. This protection addresses multiple security threats, including adversaries in network communications, multi-tenant vulnerabilities during computation on servers, and insider attacks on stored data.

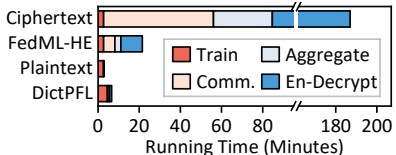

Figure 2: Time breakdown for training a ViT on GTSRB.

While platforms such as IBM FL [39] and Nvidia FLARE [35] have explored integrating HE into FL, they fail to address its significant overhead. As shown in Figure 2, HE-related operations dominate training time, and ciphertext expansion substantially increases communication costs. Reducing HE's computational and communication overhead is key to realizing its practical benefits in FL.

### 2.2 Efficient HE-based Federated Learning

Many efforts have been made to improve the efficiency of HE-based FL. These approaches can be broadly classified into two categories: encryption-scheme optimization and algorithmic optimization.

Quantization [21, 40–43] and packing [21, 44, 45] are widely studied techniques within the realm of encryption-scheme optimization for HE-based FL. Quantization reduces communication costs by converting high-precision gradients into low-precision values, whereas packing (also referred to as batching) consolidates multiple local gradients into a single plaintext, significantly reducing the number of plaintexts that need to be encrypted and transmitted.

Algorithmic optimization involves tailoring efficient strategies based on the characteristics of the machine learning model, and our DictPFL falls into this category. The state-of-the-art work, FedML-HE [31] proposes to selectively encrypt the gradients based on privacy-sensitive scores, i.e., *Select-and-Encrypt (SaE)*, as shown in Figure 1 (a). However, it suffers from several critical limitations. First, privacy-sensitive scores are computed once before training and remain static throughout the training process. This static approach fails to account for how weight sensitivity changes during training, because weights classified as non-sensitive on the initialized model may later become critical for privacy protection. Most critically, it cannot ensure complete privacy protection. Since only the gradients of selected parameters are encrypted, the remaining gradients are transmitted in plaintext, leading to inevitable information leakage and making it impossible to guarantee privacy protection regardless of which gradients are selected for encryption. Additionally, as illustrated in Figure 2, although FedML-HE substantially reduces the communication overhead and HE operations

(including aggregation, encryption, and decryption) by a factor of ten when only the top 10% of sensitive parameters are encrypted, these overheads induced by ciphertexts are still primary bottlenecks in the training process. In contrast, our DictPFL effectively reduces HE-related overhead and achieves efficiency comparable to non-private plaintext FL.

## 2.3 Motivation

As illustrated in Figure 2, communication and computation overheads caused by ciphertexts become the main bottleneck in HE-based FL. Although the state-of-the-art FedML-HE [31] attempts to improve efficiency by selectively omitting encryption for partial parameters, it not only compromises privacy but also continues to struggle with significant HE-induced communication and computation overheads. To achieve higher efficiency without sacrificing privacy, we focus on reducing the total number of trainable parameters. Guided by this principle, we propose DictPFL, which employs two strategies: *Decompose-for-Partial-Encrypt (DePE)* (Sec. 4.1) to decompose gradients and *Prune-for-Minimum-Encrypt (PrME)* (Sec. 4.2) to prune the gradients of parameters with minimal updates.

## 3 Preliminaries

### 3.1 System Overview

Same with FedML-HE [31], the workflow of HE-based privacy-preserving federated learning begins with clients using a trusted key authority to generate a public-secret HE key pair. During each training iteration: (1) clients compute local gradients; (2) these gradients are encrypted with the public key and transmitted to the server; (3) the server aggregates the encrypted gradients; and (4) the aggregated ciphertext is broadcast back to the clients, who decrypt it using their secret keys and update their local models with the decrypted result.

### 3.2 Threat Model

We consider a semi-honest adversary $\mathcal{A}$ that may corrupt the server, which is the same as the setting of FedML-HE [31]. While $\mathcal{A}$ follows the protocol, it attempts to infer private information from benign participants. Security guarantees ensure $\mathcal{A}$ learns no information from the data of clients.

## 4 DictPFL

DictPFL consists of two modules: *Decompose-for-Partial-Encrypt (DePE)* and *Prune-for-Minimum-Encrypt (PrME)*. *DePE* decomposes model weights into a fixed global dictionary $D$ and a trainable lookup table $T$. Only the encrypted gradients of $T$ are transmitted and aggregated, whereas $D$ remains identical across clients and never leaves local devices. *PrME* further reduces encryption cost by pruning parameters with persistently small gradients, using shared historical statistics to ensure consistent pruning across clients. Together, these two modules ensure that all transmitted gradients are encrypted and all unencrypted ones remain strictly local, while significantly reducing the number of ciphertexts exchanged between clients and the server to achieve high efficiency without compromising privacy.

### 4.1 Decompose-for-Partial-Encrypt (DePE)

Model weight decomposition, representing a weight matrix $W$ as a linear combination of vectors from a compact dictionary $D$ and a sparse lookup table $T$, is a proven strategy for parameter reduction in inference [46–48]. The key insight lies in reducing the inherent redundancy in weight parameters: corre-lated parameters can be represented as sparse linear combinations of a dictionary of vectors. We adapt

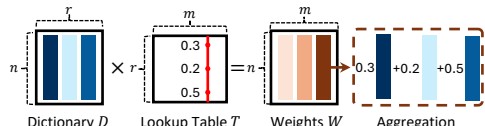

Figure 3: Representing the weight matrix $W$ with dictionary $D$ and lookup table $T$.

this principle to HE-based FL, where reducing the dimensionality of trainable parameters directly minimizes the number of ciphertexts.

**Constructing $W$ with $D$ and $T$.** Figure 3 demonstrates the construction of the weight matrix $W \in \mathbb{R}^{n \times m}$ using a dictionary $D \in \mathbb{R}^{n \times r}$ and a lookup table $T \in \mathbb{R}^{r \times m}$. Each column vector $W[:][i]$ of $W$ is derived through a linear combination of the $r$ vectors in $D$, weighted by the corresponding scalars in the $i$-th column of $T$, denoted $T[:][i]$. This process is formally expressed by:

$$W[:][i] = \sum_{k=0}^{r} D[:][k] \cdot T[k][i] \tag{1}$$

Take Figure 3 as an example. Given $r = 3$ and the $i$-th column of $T$ as $[0.3, 0.2, 0.5]$, the corresponding $i$-th column of weights in $W$ can be represented as $W[:][i] = 0.3 \cdot D[:][0] + 0.2 \cdot D[:][1] + 0.5 \cdot D[:][2]$. By reducing $r$, the dictionary size, we effectively decrease the number of trainable parameters and thus reduce the communication overhead associated with ciphertexts.

**Factorization of Dictionary and Lookup Tables.** To ensure that the dictionary $D$ contains critical and generalizable weight vectors while reducing parameter redundancy, we employ a truncated SVD factorization to decompose the weights to be trained, i.e., $W_0$, which has dimensions $n \times m$, into a smaller dictionary $D$ and a lookup table $T'$. Specifically, $W_0$ is approximated as $U_r \Sigma_r V_r^\top$, where $U_r$, $\Sigma_r$, and $V_r^\top$ correspond to the top-$r$ singular values and vectors, thus reducing the dimensionality to $n \times r$ for $D$ and $r \times m$ for $T'$,

$$W_0 \approx U_r \Sigma_r V_r^\top \tag{2}$$

$$D, T' = SVD(W_0, r) = U_r \Sigma_r, V_r^\top \tag{3}$$

DePE initializes $D$ as $U_r \Sigma_r$ and $T'$ as $V_r^\top$ according to Equation 3. However, directly freezing $D$ and training $T'$ can lead to suboptimal performance due to the information loss inherent in SVD truncation, particularly when $r$ is much smaller than $m$ or $n$. To counteract this, we retain the original weight $W_0$ and initialize $T$ by zeroing out $T'$. This strategy allows for the construction of $W$ as $W_0 + D \cdot T$, with $D$ remaining static and shared among all clients, while $T$ is updated locally and aggregated on the server. By selecting a smaller $r$, we significantly reduce the communication overhead for encrypted parameters, as encryption is only required for the $r \times m$ entries in $T$.

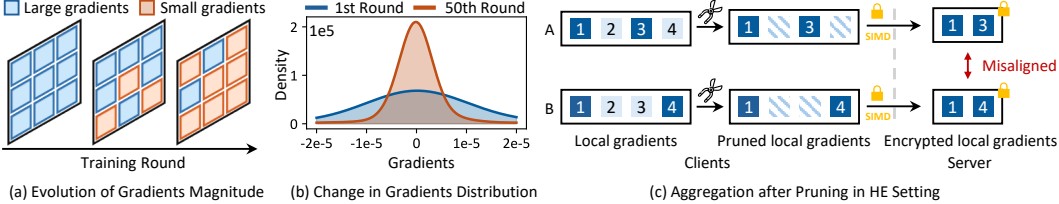

Figure 4: (a) As training progresses, parameters that initially have large gradients gradually transition to having smaller ones. (b) Concurrently, the number of parameters with substantial gradients decreases significantly. (c) An example of failed aggregation caused by different pruning locations on clients A and B.

## 4.2 Prune-for-Minimum-Encrypt (PrME)

As DePE training progresses, the number of parameters with large gradients gradually declines, as shown in Figure 4 (a). By the 50-th training round, only a small subset of parameters still exhibit gradients exceeding $10^{-5}$, as shown in Figure 4 (b). Encrypting and transmitting all gradients to the server for aggregation, including those of parameters that no longer change significantly, introduces unnecessary redundancy. By enabling clients to upload only substantial gradients, communication overhead can be dramatically reduced.

Existing gradients pruning methods in plaintext federated learning involve clients independently pruning their smallest local gradients before transmission to the server for aggregation. Since clients possess different local gradients, they may prune parameters at different positions, necessitating the sharing of pruning indices with the server to ensure proper aggregation. However, implementing such methods to HE-based federated learning presents two fundamental challenges. First, indices must be encrypted to protect privacy, while encrypted indices force the server to perform non-linear operations (e.g., comparing encrypted indices to match) alongside linear operations (e.g., aggregation), a hybrid workflow that incurs prohibitive computational overhead [49, 50]. Second, the SIMD batching

mechanism, which packs multiple plaintext gradients into several slots of a single ciphertext, renders index-specific operations infeasible. Since HE aggregation occurs slot-wise, gradients occupying the same slot across clients are combined automatically, regardless of their indices.

Figure 4 (c) illustrates the above challenges of pruned HE aggregation. Consider a scenario where client A encrypts and uploads gradients from positions 1 and 3, while client B encrypts and uploads gradients from positions 1 and 4. The server cannot perform correct aggregation because the ciphertext slots are misaligned, and the encryption prevents any coordination or realignment of the gradients. To ensure consistent gradient pruning across clients, they require an identical metric for determining which gradients to prune. The optimal approach would involve clients pruning their local gradients based on current round global gradients. However, clients cannot access the current round global gradients until after sharing their complete local gradients with the server for aggregation. This creates a dilemma: clients cannot prune independently as it leads to inconsistencies, nor can they rely on global gradients to coordinate pruning.

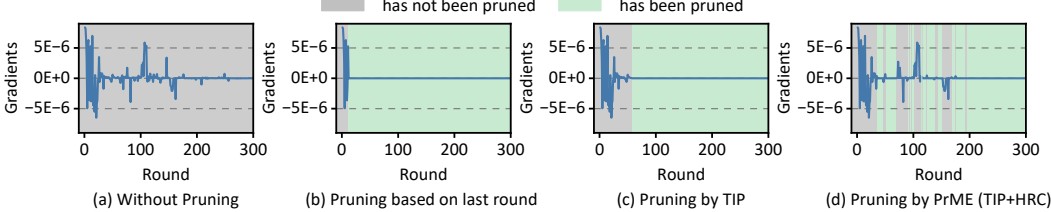

(a) Without Pruning    (b) Pruning based on last round    (c) Pruning by TIP    (d) Pruning by PrME (TIP+HRC)

Figure 5: Evolution of a parameter's global gradients under different pruning strategies. Green background indicates the parameter is pruned (excluded from aggregation), while gray background indicates the opposite. Larger green areas reflect more overhead reduction. Closer alignment of gradient trends with the baseline (a) signifies preserved convergence performance.

**Temporal Inactivity Pruning (TIP).** To resolve this dilemma, clients require a shared pruning metric that is independent of the current round's global gradients. A straightforward solution is to base pruning decisions on the last round's global gradients, which are identical across clients and accessible before aggregation. Specifically, clients prune local gradients corresponding to parameters with the smallest $s\%$ magnitudes from the prior global gradients. However, parameters showing minimal activity in one round may experience significant updates in subsequent rounds, leading to unintended removal if pruning decisions rely exclusively on last round gradients. For instance, as illustrated in Figure 5 (b), the parameter with a small gradient magnitude in an earlier round may be pruned, despite its gradient resurgence in later rounds, as indicated in Figure 5 (a).

To mitigate the influence of transient fluctuations and retain critical gradients, we introduce a temporal windowing strategy that leverages information from the previous $\tau$ consecutive rounds. Clients identify parameters whose gradients fall within the smallest $s\%$ across all $\tau$ rounds (referred to as pruning patience). Formally, the pruning mask for parameter $w_i$ at round $t$ is defined as:

$$M_{i,t} = \begin{cases} 0 & \text{if } \sum_{k=1}^{\tau} \mathbf{1}\left(|\delta w_{i,t-k}| < \theta_{s,t-k}\right) = \tau \\ 1 & \text{otherwise} \end{cases} \tag{4}$$

Here, $M_{i,t} = 0$ indicates that the local gradient of $w_i$ is pruned, while $M_{i,t} = 1$ retains its local gradient for aggregation. The $\delta w_{i,t-k}$ denotes the global gradient of parameter $w_i$ at round $t - k$, and $\mathbf{1}$ is the indicator function. The threshold $\theta_{s,t-k}$ dynamically adapts as the $(100\text{-}s)$-th percentile of $|\delta w_{i,t-k}|$. As shown in Figure 5 (c), the pruning is postponed to a later round when gradients exhibit more stable behavior, thereby preserving gradients that regain significance after initially being considered for pruning.

**Holistic Reactivation Correction (HRC).** Although TIP reduces communication overhead while preventing premature pruning by chance, it still has an inherent limitation: once a parameter is pruned, its local gradients no longer participate in aggregation. Consequently, its global gradient magnitudes remain zero in subsequent rounds, effectively excluding it permanently. This irreversible pruning can hinder training convergence, as parameters with substantial gradients in later rounds may no longer be updated. For example, in Figure 5 (a), the example parameter may have siginificant gradient magnitudes even after the 100-th round.

To mitigate the performance loss caused by irreversible pruning, we propose a dynamic reactivation scheme, Holistic Reactivation Correction (HRC). Instead of permanently excluding pruned parameters, HRC assigns each pruned parameter $w_i$ a reactivation probability $p_i$, which is dynamically adjusted based on its aggregated global gradients $\delta w_{i,t}$ after reactivation:

$$p_i[t+1] = \begin{cases} p_i[t] \times \beta & \text{if } |\delta w_{i,t}| < \theta_{s,t} \\ \min\left(p_i[t]/\beta, 1\right) & \text{otherwise} \end{cases} \tag{5}$$

Here, $\beta$ is a decay factor less than 1. When a pruned parameter is reactivated, the client uploads its *accumulated* local gradients since the pruning round for aggregation and gets the current round's global gradients $\delta w_{i,t}$. This approach preserves small gradients that, while individually minor, can meaningfully accumulate over time, rather than discarding these gradients, maintaining them locally for future aggregation helps convergence. If $|\delta w_{i,t}| < \theta_{s,t}$, indicating that the parameter's cumulative global gradients remain small even after reactivation, the reactivation probability $p_i$ decreases, discouraging further reactivation. Conversely, if $|\delta w_{i,t}| \geq \theta_{s,t}$, $p_i$ increases, encouraging the update of this parameter to rejoin aggregation. This adaptive mechanism mitigates information loss from premature pruning by flexibly adjusting the likelihood of reactivation. Although HRC introduces some uncertainty, consistency across clients can be easily maintained by preserving a shared random seed for the pruning mask. Notably, our PrME does not need to share pruning mask to server, eliminating the risk of attacks via plaintext mask, e.g., inferring sensitive patterns from pruned parameter locations.

## 5 Experimental Methodology

**Datasets.** We conduct experiments on three image classification tasks: CIFAR-10 [51], GTSRB [52], and Diabetic Retinopathy [53], as well as AG's News [54] for sentence classification and Meta-MathQA [55] for text generation. The experiments are performed under varying levels of data heterogeneity and different numbers of clients. We generate homogeneous data splits by randomly assigning training examples to individual clients without replacement. For heterogeneous settings, we simulate data heterogeneity by sampling the label ratios from a Dirichlet distribution with a symmetric parameter, following the [56]. In both settings, each client holds the same number of samples, following [57].

**Models.** We perform DictPFL on multiple prevalent transformer-based models including, ViT [58] designed for image recognition, BERT [59], and TinyLlama [60] for natural language processing.

**Baselines.** We compare DictPFL with three baselines: FedHE-Full [35], which trains the entire model and encrypts all gradients; FedHE-Top2, fine-tuning only the last two layers; and FedHE-ML [31], which encrypts a subset of gradients (10% unless specified otherwise) while leaving the rest in plaintext.

**Evaluation Metrics.** We assess the efficacy of our proposed DictPFL by comparing its communication overhead, training time, and model accuracy against existing methods. For privacy evaluation, we compare DictPFL with FedML-HE [31] in terms of potential privacy leakage. We utilize recovered image similarity scores derived from $1 - \text{LPIPS}$, where the Learned Perceptual Image Patch Similarity (LPIPS) [61] measures discrepancies between reconstructed and original images. Therefore, higher scores indicate greater similarity and consequently, higher privacy risks.

**Hyperparameters.** Unless otherwise specified, we set the dictionary size $r$ to 4, the pruning ratio $s\%$ to 70%, the pruning patience $\tau$ to 3, and the reactivation probability scaler $\beta$ to 0.2. Detailed analyses of these hyperparameters are provided in Section 6.2.

**HE Implementation.** We adopt the CKKS homomorphic encryption scheme with bootstrapping [62–64], implemented via OpenFHE [65]. The scheme is configured for 128-bit security following the Homomorphic Encryption Standard [66], with a cyclotomic ring dimension of $N = 2^{16}$, ciphertext modulus of 1555 bits, and multiplicative depth $L = 12$. Each ciphertext contains $N/2 = 32,768$ slots, enabling parallelized SIMD operations [67]. Data encoding follows the approach in [68]. All experiments were conducted on an AMD Ryzen Threadripper PRO 3955WX processor (2.2 GHz) with 125 GB of memory.

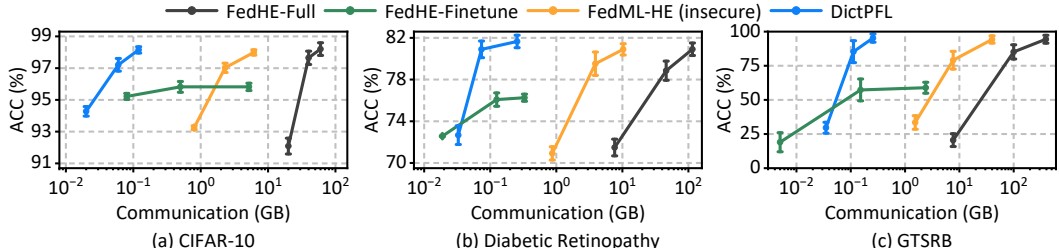

Figure 6: Efficiency comparison of different federated learning frameworks, in terms of accuracy versus communication overhead on three datasets using the ViT model. Higher efficiency is indicated by higher accuracy for the same communication or achieving the same accuracy with less communication, as shown by lines closer to the upper left corner. Communication is quantified by the total amount of data exchanged, including both plaintexts and ciphertexts, training iterations.

# 6 Results

## 6.1 Main Results

**Comparison with Existing Works.** To demonstrate DictPFL's effectiveness, we compare it with other HE-based FL frameworks on the CIFAR-10, Diabetic Retinopathy, and GTSRB datasets using the ViT-16 model within a 3-client homogeneous setting. All experiments are conducted on the same pre-trained model to ensure a fair comparison. Figure 6 provides an overall comparison. Notably, DictPFL significantly and consistently reduces communication overhead compared to the baselines without sacrificing accuracy. Specifically, FedHE-Full has the highest communication. FedHE-Top2, which fine-tunes only the last two layers, shows reduced overhead but underperforms, because freezing most layers limits learning capacity, particularly on datasets that diverge from those used in pre-training. For instance, it achieves only 58.9% accuracy on GTSRB versus DictPFL's 95.27%.

DictPFL achieves a 98.3% average reduction in communication overhead compared to the state-of-the-art FedML-HE (encrypting 10%), while maintaining the same level of accuracy. Although FedML-HE also reduces communication costs, it does so at the expense of privacy by exposing part of the gradients in plaintext. DictPFL, on the other hand, fully preserves privacy. This is further demonstrated in Figure 7 (a), which highlights the vulnerability of FedML-HE to state-of-the-art gradient inversion attacks [69]. Notably, DictPFL can prevent such privacy leakage for any data type, not only for vision tasks.

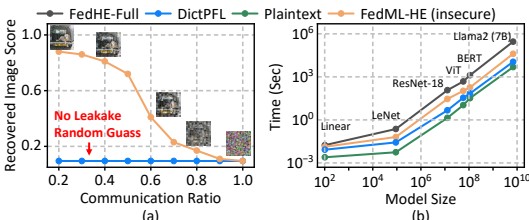

Figure 7: (a) Gradient inversion attacks against FedML-HE and DictPFL. The communication ratio is the communication overhead relative to encrypting the full-size model gradients in FedHE-Full. (b) Communication overhead of DictPFL and the baselines on models of different sizes.

In addition to ViT, we evaluate several other models, as shown in Figure 7 (b). The results show that DictPFL consistently outperforms the baselines across models of different scales. Compared with the fully encrypted baseline FedHE-Full, DictPFL reduces communication by 402 to 748 times and accelerates training by 28 to 65 times. It also outperforms the selectively encrypted baseline FedML-HE by reducing overhead by 51 to 155 times and speeding up training by 4 to 19 times.

**Breakdown Analysis.** In Figure 8, we break down the training time for various HE-based FL frameworks under both LAN and WAN settings. In FedHE-Full, where all gradients are encrypted, communication and ciphertext-related operations (encryption, decryption, and aggregation) dominate the training time. FedHE-Top2 reduces communication and ciphertext-related operations by fine-tuning the last two layers, but this comes at the cost of reduced accuracy, achieving only 58.9%. On the contrary, our proposed DePE and PrME techniques significantly reduce the number of ciphertexts, resulting in a total training time that is 1 to 2 orders of magnitude lower than that of other baselines while maintaining a comparable level of accuracy.

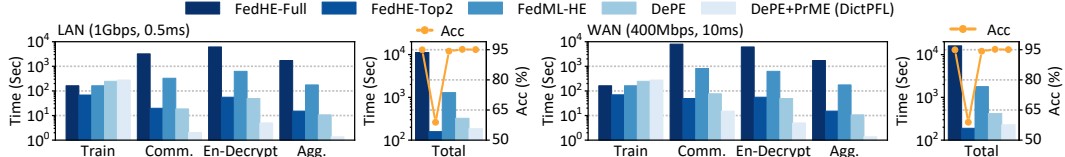

Figure 8: Training time breakdown of ViT on GTSRB under LAN and WAN settings.

## 6.2 Ablation Study

In this section, we explore the design space of DictPFL and study the impact of various settings on its performance. Unless otherwise specified, all experiments are conducted using the Diabetic Retinopathy dataset within a 3-client homogeneous setting within 10 rounds, and follows the default hyperparameter settings detailed in Section 5.

**Hyperparameters of DePE.** The dictionary size is a crucial hyperparameter in our DePE. A larger dictionary captures more comprehensive representations of gradients, enhancing accuracy but increasing overhead. As shown in Table 1, even a small dictionary with $r = 4$ achieves commendable training performance, e.g., an accuracy of $81.99\%$, close to the $82.74\%$ achieved by FedHE-Full. This efficacy stems from the dictionary's ability to retain essential information corresponding to the largest singular values.



Table 1: Ablation on dictionary size $r$.

| $r$ | Accuracy (%) ↑ | Comm. (GB) ↓ | Time (min) ↓ |
|---|---|---|---|
| 2 | $74.26_{\pm 0.5}$ | 0.046 | $6.11_{\pm 0.1}$ |
| 4 | $81.99_{\pm 0.4}$ | 0.088 | $6.23_{\pm 0.1}$ |
| 8 | $82.67_{\pm 0.2}$ | 0.160 | $6.42_{\pm 0.2}$ |
| 16 | $82.71_{\pm 0.2}$ | 0.332 | $7.27_{\pm 0.1}$ |

Table 2: Ablation on pruning patience $\tau$.

| $\tau$ | Accuracy (%) ↑ | Comm. (GB) ↓ | Time (min) ↓ |
|---|---|---|---|
| 1 | $80.55_{\pm 0.6}$ | 0.001 | $6.26_{\pm 0.1}$ |
| 3 | $82.29_{\pm 0.3}$ | 0.003 | $6.36_{\pm 0.1}$ |
| 5 | $82.67_{\pm 0.2}$ | 0.160 | $6.42_{\pm 0.2}$ |
| 10 | $82.77_{\pm 0.3}$ | 0.474 | $6.92_{\pm 0.1}$ |



**Hyperparameters of PrME.** We explore the impact of pruning ratio $s\%$ and pruning patience $\tau$ in PrME. A higher $s\%$ results in more minor gradients being pruned, whereas a lower value preserves them. As shown in Figure 9, without PrME (prune $0\%$), training converges rapidly within 10 rounds, but each round incurs the highest communication cost. Pruning $70\%$ drastically reduces communication overhead but significantly affects accuracy. By contrast, pruning $20\%$ preserves accuracy but results in far less communication reduction compared to the $70\%$ pruning scenario. Notably, with our HRC reactivation scheme, prematurely pruned gradients in earlier rounds can be selectively reintroduced in later rounds. This enables the model to achieve accuracy similar to the $20\%$ pruning scenario while achieving the communication efficiency of the $70\%$ pruning ratio.

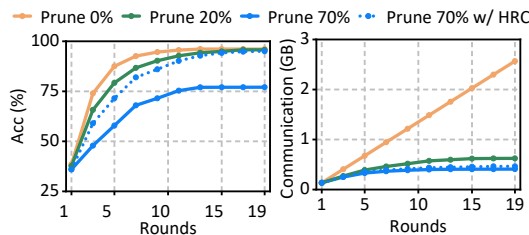

Figure 9: Ablation on the pruning ratio.

Table 2 studies different pruning patience $\tau$. Higher $\tau$ values delay the pruning of gradients, reducing accuracy degradation but limiting communication reduction. Notably, setting $\tau = 3$ already results in a small accuracy loss. This resilience can be attributed to our HRC, which mitigates the impact on accuracy by reintroducing pruned gradients, effectively correcting errors over time.

Table 4 in Appendix A.2 showcases that our PrME works well under various reactivation probability scalers $\beta$. For different numbers of clients and heterogeneous levels, we show the results in Appendix A.3 and A.4, which show that DictPFL performs well across different client scales and heterogeneous settings.

## 6.3 Other Experiments

The results for text tasks and large language models, including classification and generation tasks are in Appendix A.5. DictPFL outperforms all the baselines on language tasks. In Appendix A.7, we compare DictPFL with other non-HE based FL.

# 7  Conclusion

In this work, we present DictPFL, a novel framework for efficient HE-based FL. To address the prohibitive ciphertext-related overhead and eliminate information leakage, we propose *Decompose-for-Partial-Encrypt (DePE)*, which decomposes model weights into a static dictionary and a trainable lookup table. Only the small lookup table is encrypted and shared for aggregation, while the dictionary is never transmitted. To further improve communication efficiency, we propose *Prune-for-Minimum-Encrypt (PrME)*, which prunes gradients based on their long-term activity to minimize redundant ciphertext operations. Compared with the fully encrypted baseline, DictPFL accelerates training by up to $65\times$ and outperforms the selectively encrypted FedML-HE by up to $19\times$ while maintaining accuracy and fully eliminating privacy risks from partial plaintext gradient transmission, achieving a runtime only $2\times$ that of plaintext FL.

# 8  Discussion

**Broader Impact.** The paper introduces DictFPL, a method designed to reduce the computational and communication overheads associated with protecting federated learning shared weights using homomorphic encryption. This approach enhances privacy protections without compromising accuracy, making it a more feasible solution for large-scale, real-world applications. By ensuring that sensitive weights remains private, DictFPL can accelerate the adoption of federated learning across industries such as healthcare, finance, and beyond, while fostering trust in AI systems and promoting global data privacy.

**Limitations.** Future work could explore broader scenarios, such as cross-device FL with constrained client resources or non-transformer model families. Moreover, since DictPFL employs a **fixed** shared dictionary, extending it to a **dynamic** dictionary design could enhance adaptability in highly heterogeneous client environments and improve model personalization.

# 9  Acknowledgement

This work was supported in part by NSF CSR-2413232. Any opinions, findings and conclusions or recommendations expressed in this material are those of the authors and do not necessarily reflect the views of grant agencies or their contractors.

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

# Appendix

## A   More Experiments

### A.1   Comparison without pre-trained weights

As shown in Table 3, even without using pre-trained weights, DictPFL achieves the highest accuracy among all methods, reaching 95.06%, compared to 94.17% for FedHE-FULL and 94.99% for FedML-HE. More importantly, DictPFL offers substantial efficiency gains: the total communication cost is reduced to 0.51 GB, while FedHE-FULL and FedML-HE require 720.72 GB and 73.62 GB, respectively. In terms of training time, DictPFL completes in just 11.8 minutes, far less than 294.6 minutes for FedHE-FULL and 56.7 minutes for FedML-HE.

Table 3: Comparison with baselines on without pre-trained weights.

|  | Acc. (%) $\uparrow$ | Comm. (GB) $\downarrow$ | Time (min) $\downarrow$ |
|---|---|---|---|
| FedHE-FULL | 94.17 | 720.72 | 294.6 |
| FedML-HE | 94.99 | 73.62 | 56.7 |
| DictPFL (ours) | 95.06 | 0.51 | 11.8 |

### A.2   Different reactivation probability scale $\beta$

Table 4 studies different reactivation probability scalers $\beta$. The result showcase the our PrME works well under different $\beta$.

Table 4: Ablation study on $\beta$ under $s\% = 70\%$ and $\tau = 3$.

| $\beta$ | Accuracy (%) $\uparrow$ | Comm. (GB) $\downarrow$ | Time (min) $\downarrow$ |
|---|---|---|---|
| 0.2 | $82.29_{\pm 0.3}$ | 0.003 | $6.36_{\pm 0.1}$ |
| 0.5 | $82.37_{\pm 0.3}$ | 0.007 | $6.36_{\pm 0.1}$ |
| 0.8 | $82.55_{\pm 0.2}$ | 0.031 | $6.39_{\pm 0.2}$ |

### A.3   Different Number of Clients

We assess the performance of DictPFL in environments with varying numbers of clients. The findings, presented in Table 5, demonstrate that DictPFL performs effectively and consistently across settings with different client counts.

Table 5: The results of DictPFL under client numbers.

| Clients | Accuracy (%) $\uparrow$ | Comm. (GB) $\downarrow$ | Time (min) $\downarrow$ |
|---|---|---|---|
| 3 | 82.67 | 0.160 | 6.42 |
| 5 | 82.64 | 0.092 | 3.70 |
| 10 | 81.94 | 0.046 | 1.85 |
| 20 | 81.82 | 0.041 | 0.93 |
| 50 | 80.42 | 0.041 | 0.75 |
| 200 | 80.56 | 0.041 | 1.96 |

### A.4   Different Heterogeneous Level

Unsurprisingly, DictPFL performs better in homogeneous settings than in heterogeneous settings. As the table 6 shows, we evaluated DictPFL in various heterogeneous settings under different Dirichlet distributions from 0.3 to 0.9 and compared it with a homogeneous setting. The results indicate that DictPFL's performance remains stable across different heterogeneous dataset splits. Specifically, a smaller $\alpha$ (more heterogeneous) requires more communication size and training time to achieve comparable accuracy to a larger $\alpha$ (less heterogeneous).

Table 6: The results under different heterogeneous settings.

| $\alpha$ | Accuracy (%) ↑ | Comm. (GB) ↓ | Time (min) ↓ |
|---|---|---|---|
| 0.3 | $79.62_{\pm 0.4}$ | 0.103 | $6.22_{\pm 0.2}$ |
| 0.6 | $80.28_{\pm 0.2}$ | 0.145 | $6.44_{\pm 0.1}$ |
| 0.9 | $82.06_{\pm 0.3}$ | 0.151 | $6.45_{\pm 0.2}$ |
| $\infty$ | $82.67_{\pm 0.2}$ | 0.160 | $6.42_{\pm 0.2}$ |

## A.5 Performance on NLP tasks.

Table 7 shows that DictPFL significantly improves efficiency in both sentence classification and generation (instruction tuning) tasks. For the generation task, we train on the MetaMathQA [55] dataset and evaluate on GSM8K [70], focusing on mathematical reasoning. These gains are especially pronounced in larger models, where DictPFL reduces training time by 99.4% percent for TinyLlama and 96.1% percent for BERT. This improvement stems from the high cost of ciphertext operations in larger models, making DictPFL's optimizations more impactful.

Table 7: Comparison with baselines on TinyLlama and BERT.

| | Methods | Acc. (%) ↑ | Comm. ↓ | Time ↓ |
|---|---|---|---|---|
| TinyLlama-MetaMathQA | FedHE-Full | 45.86 | 30.0 TB | 214.2 h |
| | FedHE-FT | 6.92 | 2.4 TB | 17.9 h |
| | FedML-HE | 45.86 | 3.0 TB | 22.6 h |
| | DictPFL (ours) | 45.93 | 0.3 TB | 1.3 h |
| BERT-AgNews | FedHE-Full | 91.38 | 137.2 GB | 342.6 m |
| | FedHE-FT | 90.05 | 17.5 GB | 47.9 m |
| | FedML-HE | 91.38 | 13.7 GB | 32.8 m |
| | DictPFL (ours) | 91.24 | 4.8 GB | 13.4 m |

## A.6 Comparision with Non-HE based FL

We compare DictPFL with Secure Aggregation [25] by training a ViT model on the Diabetic Retinopathy dataset under a 3-client LAN setting (1 Gbps, 0.5 ms). Secure Aggregation increases training time from 257.6 s to 363.2 s, while DictPFL achieves the same 82.7% accuracy in 385.2 s. This demonstrates that HE-based FL with DictPFL is practically efficient, with much lower overhead than commonly assumed.

## A.7 Combination with Existing Quantization Techniques

Algorithmic optimization directly reduces gradient redundancy without sacrificing accuracy, whereas quantization and packing often lead to accuracy degradation. Moreover, their improvements are limited and easily saturate [71]. More importantly, these optimization approaches are orthogonal and can be combined—by first reducing gradients through algorithmic optimization and then applying quantization or packing, overall communication cost can be further minimized. Here we perform experiments (3-client ViT on CIFAR-10, to compare DictPFL (algorithmic optimization) and AdaptiveBatchHE [41] (packing optimization), as shown in the table below. It reveals that DictPFL outperforms AdaptiveBatchHE in both efficiency and accuracy and combining them will further reduce communication overhead.

Table 8: Comparison of accuracy and communication cost among HE-based frameworks.

| | Accuracy (%) ↑ | Communication (GB) ↓ |
|---|---|---|
| FedHE-FULL | 98.2 | 60.14 |
| AdaptiveBatchHE | 96.7 | 5.41 |
| DictPFL (ours) | 98.2 | 0.43 |
| DictPFL + AdaptiveBatchHE | 96.6 | 0.0872 |

## A.8 Client Personality Preservation

DictPFL preserves the personality of each client based on our HRC mechanism. Specifically, while pruning is guided by the magnitude of global gradients, the HRC mechanism allows each client to upload accumulated local gradients for parameters that are reactivated, even if they were previously pruned due to low global gradient magnitudes. This ensures that important client-specific significant gradients are not lost: whenever such parameters are reactivated again, clients contribute their accumulated local gradients.

In Table 9, we evaluate DictPFL under various degrees of data heterogeneity by adjusting the Dirichlet factor $\alpha$. While greater heterogeneity (lower $\alpha$) increases training time and communication overhead, DictPFL consistently maintains strong performance.

To further demonstrate the effect of HRC's accumulative gradient sharing mechanism, we compare "accumulative gradient sharing" versus "non-accumulative sharing," measuring the resulting accuracy under comparable training time. Our results (3-client ViT on Diabetic Retinopathy, $r = 4$, $s = 0.7$, $\tau = 3$, $\beta = 0.2$) show that omitting accumulated gradients notably reduces accuracy, particularly in more heterogeneous settings—because discarding small but meaningful gradients impairs learning for clients with diverse data. Overall, these results highlight that DictPFL achieves robust performance across a wide range of data distributions.

Table 9: Effect of accumulative gradient sharing under different Dirichlet factors $\alpha$.

| $\alpha$ | Accumulative gradient sharing (%) | Non-accumulative sharing (%) |
|---|---|---|
| 0.3 | 79.62 | 74.26 |
| 0.6 | 80.28 | 76.13 |
| 0.9 | 82.06 | 80.45 |

# B  Analysis on FedML-HE [31]

FedML-HE trades security for efficiency, and this trade-off persists regardless of whether sensitivity is dynamically recalculated. While dynamic recalculation can enhance security, it incurs substantial computational overhead to achieve an empirical $0\%$ attack success rate. Because recalculating sensitivity scores requires each client to perform a forward pass on the training dataset and share encrypted sensitivity values for secure aggregation, it introduces overhead comparable to the original training round and HE aggregation step.

Our experiments (3-client ViT on CIFAR-10, encrypt $10\%$) with varying recalculation frequencies (i.e., recalculating every $K$ rounds) show that more frequent updates do improve privacy, but at the cost of significantly reduced efficiency. Even under these settings, FedML-HE still cannot achieve the strong privacy guarantees or efficiency of DictPFL.

Table 10: Effect of dynamic sensitivity recalculation in FedML-HE.

| Method | Accuracy (%) ↑ | Communication (GB) ↓ | Attack Success Rate (LPIPS) ↓ |
|---|---|---|---|
| FedHE-Full | 98.17 | 60.14 | 0.00 |
| FedML-HE (K=1) | 98.16 | 61.07 | 0.00 |
| FedML-HE (K=2) | 98.16 | 54.28 | 0.092 |
| FedML-HE (K=5) | 98.16 | 30.8 | 0.309 |
| FedML-HE (K=10) | 98.16 | 14.2 | 0.788 |
| DictPFL (ours) | 98.15 | 0.43 | 0.00 |

