# OpenReview forum: "DictPFL: Efficient and Private Federated Learning on Encrypted Gradients"
_NeurIPS.cc/2025/Conference — NeurIPS 2025 poster_

### Official Review · Reviewer_1kTE · 2025-06-27

**Clarity:** 2
**Significance:** 2
**Originality:** 2
**Rating:** 4
**Confidence:** 2

**Summary:**

This paper introduces a novel method DictPFL to reduce the communication overhead while ensuring the privacy in HE-based FL. This goal is achieved by two modules designed, i.e., the decompose-for-partial-encrypt and prune-for-minimum-encrypt. The proposed DictPFL method achieves decent performance on different tasks, i.e., image recognition, text classification, and text generation.

**Questions:**

Please refer to the concerns listed in the weaknesses part.

**Ethical Concerns:**

["NO or VERY MINOR ethics concerns only"]

**Final Justification:**

Most of my concerns have been addressed in the rebuttal. As a result, I’d like to raise my score to 4.

**Limitations:**

yes

**Quality:**

2

**Strengths And Weaknesses:**

**Strengths**
+ The paper is well written and easy to read.
+ The visualizations of the experimental results are nicely presented.
+ The proposed DictPFL method significantly reduces the communication overhead in the domain of HE-based FL.

**Weaknesses**
+ The motivation for adopting algorithmic optimization rather than the quantization and packing needs to be further clarified. Is there any advantage brought by the algorithmic optimization?
+ This work appears to be some incremental improvement on FedML-HE. The contributions to the community of HE-based FL are expected to be clarified.
+ There is an identical metric for determining which gradients to prune. However, if the data distribution of the clients is different, how to ensure the personality of each client based on this design?
+ Please provide additional experimental results in terms of the convergence of the training process of DictPFL.

---

> ### Author Rebuttal · Authors · 2025-07-31
>
> We thank Reviewer 1kTE for the insightful feedback on the clarity of our writing and the quality of our visualizations. We are also pleased that the effectiveness of DictPFL in significantly reducing communication overhead in HE-based federated learning was appreciated.
>
> **Q1. The motivation for adopting algorithmic optimization rather than the quantization and packing needs to be further clarified. Is there any advantage brought by algorithmic optimization?**
>
> We thank the reviewer for the insightful question. Algorithmic optimization directly reduces gradient redundancy without sacrificing accuracy, whereas quantization and packing often lead to accuracy degradation. Moreover, their improvements are limited and easily saturate [b]. More importantly, these optimization approaches are orthogonal and can be combined—by first reducing gradients through algorithmic optimization and then applying quantization or packing, overall communication cost can be further minimized. Here we perform experiments (3-client ViT on CIFAR-10, $r=4$, $s=0.7$, $\tau=3$, $\beta=0.2$) to compare DictPFL (algorithmic optimization) and AdaptiveBatchHE [a] (packing optimization), as shown in the table below. It reveals that DictPFL outperforms AdaptiveBatchHE in both efficiency and accuracy and combining them will further reduce communication overhead.
>
> |                            | Accuracy (%)  | Communication (GB)  |
> |----------------------------|---------------|---------------------|
> | FedHE-FULL                 | 98.2          | 60.14               |
> | AdaptiveBatchHE            | 96.7          | 5.41                |
> | DictPFL                    | 98.2          | 0.43                |
> | DictPFL + AdaptiveBatchHE  | 96.6          | 0.0872             |
>
> **Q2. This work appears to be some incremental improvement on FedML-HE. The contributions to the community of HE-based FL are expected to be clarified.**
>
> DictPFL is fundamentally different from FedML-HE in both principle and design. While FedML-HE improves efficiency by leaving partial gradients unencrypted—thereby trading privacy for efficiency—DictPFL improves efficiency without compromising privacy, as all shared gradients remain fully encrypted. This core difference enables DictPFL to maintain the strong privacy guarantees of HE that can defense any attacks, while significantly reducing overhead.
>
> Notably, our results demonstrate that DictPFL incurs only a 2× computational overhead compared to plaintext FL, showing for the first time that deploying FHE in FL can be practical at scale. This represents more than an incremental improvement, offering the community a practical solution for achieving both security and efficiency in HE-based FL.
>
> **Q3. There is an identical metric for determining which gradients to prune. However, if the data distribution of the clients is different, how to ensure the personality of each client based on this design?**
>
> DictPFL preserves the personality of each client based on our HRC mechanism. Specifically, while pruning is guided by the magnitude of global gradients, the HRC mechanism allows each client to upload accumulated local gradients for parameters that are reactivated, even if they were previously pruned due to low global gradient magnitudes. This means that important client-specific significant gradients are not lost: whenever such parameters are reactivated again, clients contribute their accumulated local gradients. In Table 6, we evaluate DictPFL under various degrees of data heterogeneity by adjusting the Dirichlet factor $\alpha$. While greater heterogeneity (lower $\alpha$) increases training time and communication overhead, DictPFL consistently maintains strong performance.
>
> To further demonstrate the effect of HRC's accumulative gradient sharing mechanism, we compare “accumulative gradient sharing” versus “non-accumulative sharing,” measuring the resulting accuracy under comparable training time. Our results (3-client ViT on Diabetic Retinopathy, $r=4$, $s=0.7$, $\tau=3$, $\beta=0.2$) show that omitting accumulated gradients notably reduces accuracy, particularly in more heterogeneous settings—because discarding small but meaningful gradients impairs learning for clients with diverse data. Overall, these results highlight that DictPFL can achieve robust performance across a wide range of data distributions.
>
> | $\alpha$  | Accumulative gradient sharing  | Non-accumulative sharing  |
> |-----------|--------------------------------|---------------------------|
> | 0.3       | 79.62%                         | 74.26%                    |
> | 0.6       | 80.28%                         | 76.13%                    |
> | 0.9       | 82.06%                         | 80.45%                    |
>
> **Q4. Please provide additional experimental results in terms of the convergence of the training process of DictPFL.**
>
> In our main paper, we provided learning curve plots in Figure 9 to demonstrate DictPFL’s convergence. To further address the reviewer’s concern, we will include additional plots in the revision that show the training and test losses over training rounds, across different settings.  All experiments in the main paper were repeated over five independent runs, and we report the mean and standard deviation for key metrics such as accuracy, communication cost, and training time (see Figure 6 and Table 1 and 2). The statistical reporting primarily demonstrates the stability and reproducibility of DictPFL’s performance, the consistently low standard deviations across these metrics provide indirect evidence of robust convergence.
>
> [a] Adaptive Batch Homomorphic Encryption for Joint Federated Learning in Cross-Device Scenarios, IOT 2023.
>
> [b] HEQuant: Marrying Homomorphic Encryption and Quantization for Communication-Efficient Private Inference

---

> > ### Comment · Reviewer_1kTE · 2025-08-05
> >
> > Thanks for the detailed rebuttal. Most of my concerns have been addressed. As a result, I’d like to raise my score to 4.

---

### Official Review · Reviewer_JGjF · 2025-07-03

**Clarity:** 2
**Significance:** 3
**Originality:** 3
**Rating:** 4
**Confidence:** 2

**Summary:**

This paper introduces DictPFL for efficient HE-based federated learning. Through Decompose-for-Partial-Encrypt (DePE), which decomposes model weights into a static dictionary and a trainable lookup table, and further optimization via Prune-for-Minimum-Encrypt (PrME), DictPFL substantially reduces the transmission of encrypted gradients while preserving privacy guarantees. Experimental results demonstrate that DictPFL achieves significant efficiency across various tasks.

**Questions:**

Can the two strategies proposed in the paper, DePE and PrME, be applied to other non-HE-based federated learning methods?

The paper suffers from redundant expressions. For instance, the motivation and contributions in the abstract and introduction are not concisely condensed; the structure of other sections also differs significantly from that of conventional papers, making these sections overly verbose and difficult to follow.

The comparative methods are relatively few in number and outdated. For instance, FedHE-Full (2022), FedHE-Finetune (2023), and FedML-HE (2023) were all published before 2023.

The section structure may need adjustment.

**Ethical Concerns:**

["NO or VERY MINOR ethics concerns only"]

**Final Justification:**

The authors have addressed my questions. While their responses are relatively brief (Q2), they are barely acceptable. I choose to maintain my score.

**Limitations:**

Yes

**Paper Formatting Concerns:**

There are no major formatting issues.

**Quality:**

3

**Strengths And Weaknesses:**

The proposed method significantly improves efficiency while maintaining privacy guarantees, which is conducive to practical applications. This work is meaningful. However, the paper suffers from redundant expressions and needs further condensation and refinement. Additionally, the comparative methods are relatively few in number and outdated, and it is recommended to include more recent comparative methods.

---

> ### Author Rebuttal · Authors · 2025-07-31
>
> We sincerely thank Reviewer JGjf for recognizing the practical value of our work. We are glad that the efficiency gains and preserved privacy guarantees of our method were found meaningful and conducive to real-world deployment.
>
> **Q1. Can the two strategies proposed in the paper, DePE and PrME, be applied to other non-HE-based federated learning methods?**
>
> Thanks for the reviewer's insightful question. The proposed DePE and PrME strategies are general and applicable to non-HE-based FL, especially in communication-sensitive settings. By reducing the volume of gradient information transmitted, they help lower communication overhead regardless of encryption. DictPFL shows greater benefits in HE-based FL, where communication and computation overhead introduced by HE becomes the bottleneck (see Figure 2).
>
> In summary, while DePE and PrME are effective in non-HE scenarios, their advantages are most significant in HE-based FL, which is the focus of this work.
>
> **Q2. The paper contains redundant expressions, with an overly verbose abstract and introduction.**
>
> Thanks for the reviewer's constructive feedback. We will revise the abstract and introduction to make the motivation and contributions more concise.
>
> **Q3. The comparative methods are relatively few in number and outdated. For instance, FedHE-Full (2022), FedHE-Finetune (2023), and FedML-HE (2023) were all published before 2023.**
>
> We found that most new HE-FL approaches still use our main baselines (FedHE-Full, FedHE-Finetune, and FedML-HE) as fundamental frameworks. Many recent works (2024–2025) either build directly on these methods (e.g., [a, b], which extend FedML-HE and retain its partial encryption strategy) or explore orthogonal optimizations such as quantization and packing [c], which are compatible with our DictPFL.
>
> Our DictPFL builds upon and enhances these widely adopted baselines, ensuring a fair and representative evaluation. We will also update the Related Work section to cite and briefly discuss these recent developments [a,b,c].
>
> [a] A Selective Homomorphic Encryption Approach for Faster Privacy-Preserving Federated Learning, Arxiv 2025
>
> [b] Privacy-Preserving Federated Learning for SkinCancer Detection Using Homomorphic Encryption and Advanced Deep Learning Techniques, IJCS 2025
>
> [c] Efficient and Straggler-Resistant Homomorphic Encryption for Heterogeneous Federated Learning, INFOCOM 2025

---

> > ### Comment · Reviewer_JGjF · 2025-08-05
> >
> > The authors have addressed my questions. While their responses are relatively brief (Q2), they are barely acceptable. Hope to see improvements in the final version, and I have no further questions.

---

> ### Author Response · Authors · 2025-08-05
>
> Thank you for your positive and helpful feedback.
>
> To addressing the Q2, we will revise the paper to improve clarity and structure. Specifically, we plan the following adjustments:
>
> - Merge the first two paragraphs of the Introduction into a single concise paragraph that introduces FL privacy challenges and sets up the HE-based FL context, avoiding repetition with the Abstract.
>
> - Summarize prior work limitations at a high level in the Introduction and move detailed analysis into a new **Section 3 Motivation**, making the Introduction more focused and readable. And follow Reviewer oqcz's suggestion, using Figure 7 to illustrate the motivation.
>
> - Reorganize sections by clearly separating Related Work, Motivation, and Methodology, to enhance logical flow.
>
> We sincerely appreciate your constructive suggestions, which will guide us in refining the final version. We would be truly grateful for your support during the reviewer discussion phase.

---

### Official Review · Reviewer_oqcz · 2025-07-19

**Clarity:** 2
**Significance:** 3
**Originality:** 3
**Rating:** 5
**Confidence:** 4

**Summary:**

This paper introduces a new Homomorphic encryption (HE) method defending against gradient inversion attacks in Federated Learning (FL).
Prior HE methods encrypt every gradient update, which is time-consuming, or encrypt most sensitive gradients, which provides weaker privacy. Instead, this paper tries to encrypt every gradient that is transmitted to the server, aiming to enhance privacy and efficiency as well.

Specifically, the paper proposes the method DictPFL, including decompose-for-partial-encrypt (DePE) and Prune-for-Minimum-Encrypt (PrME). DePE decomposes model weights $W_0$ via SVD into a frozen and globally consistent dictionary $D$, which is identical across all clients and is not transmitted, and a trainable lookup table $T$, which will be encrypted and shared for aggregation, i.e., $W=W_0+DT$. PrME further minimizes the encrypted lookup table parameters $T$ on the client side, conducts consistent pruning based on the history of global gradients.

Experiments on image classification, sentence classification, and text generation demonstrate the superiority of DictPFL in terms of efficiency and effectiveness against attacks.

**Questions:**

- It is recommended to present results on performance against attacks and the accuracy of the model together under the same hyperparameters (maybe a table), and include a comparison with other baselines, for better clarity in showing the trade-off.
- For the rigorous validation of the limitation of FedML-HE (in lines 51-58), it would be better to check the vulnerability and conduct gradient inversion attacks when the top-k most sensitive parameters are encrypted, instead of based on the observation from that for magnitude-based pruning.
- The proposed method aims to protect shared gradients to enhance privacy and improve efficiency. What if FedML-HE dynamically calculates sensitivity scores to encrypt? It would be better first to check this possibility and then discuss the limitation, to provide better and clarity in supporting the motivations.
- The frozen and globally consistent dictionary $D$ is identical across all clients. Also, in line 191, need to retain the original weight $W_0$, and the dictionary may cover the coarse distribution of training data, so would it be easier for adversaries to infer gradients once they steal the dictionary?
- Currently, only one analytic attack is considered for comparison; it would be better to evaluate against different kinds of attacks, such as [51-53].
- Regarding the reactivation probability scale $\beta$, how does the method perform when $\beta=0$? From Table 4, it appears that the impact of reactivation is not significant and unnecessary.

**Ethical Concerns:**

["NO or VERY MINOR ethics concerns only"]

**Final Justification:**

I sincerely thank the authors for the detailed response; it addressed my concerns, hence I raised my score to 5.

In the revised version, it's kindly recommended to use Figure 7 to support the claim (Lines 51–58), include results about how FedML-HE dynamically calculates sensitivity scores to encrypt, and the impact of non-iid data distribution.

**Limitations:**

It is recommended to further discuss the impact of non-iid data distribution.

**Quality:**

2

**Strengths And Weaknesses:**

**Pros**
- The motivation is clear. It is important to enhance privacy while reducing the overhead for FL.
- The proposed method seems very effective and efficient, showing significant reductions in communication (402–748×) and training time (28–65×) compared to fully encrypted baselines, while maintaining accuracy.

**Cons**
- Not rigorous claims: in lines 51-58, FedML-HE encrypts only the top-k most sensitive parameters based on privacy-sensitive scores, the authors claim that this has a limitation in privacy protection because study [2] shows that 30% of gradients is enough for reconstruction. However, [2] shows that via gradient compression, where gradients with small magnitude (not important) are pruned to zero, which is magnitude-based pruning and different from the sensitive-based encryption in FedML-HE.
- Missing related works such as analytic attacks [50] and optimization-based attacks [51-53], as well as defenses such as [54].

----
[50] Fowl, L.; Geiping, J.; Czaja, W.; Goldblum, M.; and Goldstein, T. 2022. Robbing the Fed: Directly Obtaining Private Data in Federated Learning with Modified Models. ICLR.

[51] Balunovi´c, M.; Dimitrov, D. I.; Staab, R.; and Vechev, M. 2022. Bayesian Framework for Gradient Leakage. ICLR.

[52] Geiping, J.; Bauermeister, H.; Dr¨oge, H.; and Moeller, M. 2020. Inverting gradients: How easy is it to break privacy in federated learning? Advances in Neural Information Processing Systems, 33: 16937–16947.

[53] Li, Z.; Zhang, J.; Liu, L.; and Liu, J. 2022. Auditing Privacy Defenses in Federated Learning via Generative Gradient Leakage. In Proceedings of the IEEE/CVF Conference on Computer Vision and Pattern Recognition, 10132–10142.

[54] Sun, J.; Li, A.; Wang, B.; Yang, H.; Li, H.; and Chen, Y. 2021. Soteria: Provable defense against privacy leakage in federated learning from representation perspective. In Proceedings of the IEEE/CVF Conference on Computer Vision and Pattern Recognition, 9311–9319.

---

> ### Author Rebuttal · Authors · 2025-07-31
>
> We sincerely thank Reviewer oqcz for the constructive and insightful feedback. We appreciate the recognition of our clear motivation, as well as the effectiveness and efficiency of the proposed techniques.
>
> **Q1: Not rigorous claims: Lines 51–58 claim FedML-HE has limited privacy since [2] shows 30% of gradients suffice for reconstruction. However, [2] uses magnitude-based pruning, which differs from FedML-HE’s approach of encrypting top-k sensitive parameters.**
>
> Lines 51-58 were originally used to claim FedML-HE is insecure because the unencrypted gradients can leak information exploitable by attackers. Figure 7 demonstrates that when 30% of the gradients are shared unencrypted (selected by FedML-HE), the image reconstructed by attack [47] achieves a 23% similarity score. Thanks to the reviewer’s feedback, we recognize that Figure 7, rather than citation [2], should be used to support this claim. We will revise the manuscript accordingly to avoid any misunderstanding.
>
> **Q2. Missing related works such as analytic attacks [50] and optimization-based attacks [51-53], as well as defenses such as [54]. It would be better to evaluate against different kinds of attacks, such as [51-53].**
>
> Thanks for the reviewer's feedback, and we would like to clarify any misunderstanding. DictPFL is robust to all existing attacks because all shared data is encrypted. In contrast, previous method partially shares unencrypted gradients, making them vulnerable to attacks. In our paper, we use [47] as a representative attack to validate this distinction. Regarding the mentioned defense [54], the server can get the final aggregated model, whereas DictPFL prevents the server from accessing it by encrypting gradients throughout the entire process. We will clarify this point in the paper and cite all the relevant attack and defense methods accordingly.
>
> **Q3. Presenting attack performance and model accuracy together would better clarify trade-offs.**
>
> We would like to clarify any misunderstanding. DictPFL ensures security by encrypting all data shared with the server, resulting in a 0% attack success rate across all hyperparameter settings. Therefore, there is no trade-off between accuracy and security in our approach. The only trade-offs between accuracy and efficiency are presented in Figure 6.
>
> **Q4. What if FedML-HE dynamically calculates sensitivity scores to encrypt? It would be better first to check this possibility and then discuss the limitations, to provide better and clarity in supporting the motivations.**
>
> Thanks for reviewer's insightful suggestion. FedML-HE trades security for efficiency, and this trade-off persists regardless of whether sensitivity is dynamically recalculated. While dynamic recalculation can enhance security, it incurs substantial computational overhead to achieve an empirical 0% attack success rate. Because recalculating sensitivity scores requires each client to perform a forward pass on the training dataset and share encrypted sensitivity values for secure aggregation—introducing overhead comparable to original training round and HE aggregation step. Our experiments (3-client ViT on CIFAR-10, encrypt 10\%) with varying recalculation frequencies (i.e., recalculating every k-round) show that more frequent updates do improve privacy, but at the cost of significantly reduced efficiency. Even under these settings, FedML-HE still cannot achieve the strong privacy guarantees or efficiency of DictPFL.
>
> |                  | Accuracy (%)  | Communication (GB)  | Attack Success Rate (LPIPS)  |
> |------------------|---------------|---------------------|------------------------------|
> | FedHE-Full       | 98.17         | 60.14               | 0.00                         |
> | FedML-HE (K=1)   | 98.16         | 61.07               | 0.00                         |
> | FedML-HE (K=2)   | 98.16         | 54.28               | 0.092                        |
> | FedML-HE (K=5)   | 98.16         | 30.8                | 0.309                        |
> | FedML-HE (K=10)  | 98.16         | 14.2                | 0.788                        |
> | DictPFL          | 98.15         | 0.43                | 0.00                         |
>
> **Q5. Since the dictionary may reflect the coarse distribution of training data, could this make it easier for adversaries to infer gradients if the dictionary is compromised?**
>
> Under our threat model, the dictionary is never shared with the server. Even if the server somehow obtains the dictionary, it provides no advantage in inferring gradients. This is because only the lookup table is updated using private data, while the dictionary itself contains no information about gradients or client training data. The only way for the server to access gradient information would be to break the HE and decrypt the gradients, which is assumed to be computationally infeasible.
>
> **Q6. Regarding the reactivation probability scale $\beta$, how does the method perform when $\beta=0$? From Table 4, it appears that the impact of reactivation is not significant and unnecessary.**
>
> The reactivation mechanism HRC is significant and necessary. Without HRC ($\beta=0$), certain important parameters pruned in early rounds may never participate in training again, which may hinder the model's ability to achieve optimal accuracy. As shown in Figure 9, disabling HRC ($\beta=0$) results in noticeably lower accuracy with pruning 70%, i.e., 71.5%, under comparable training overhead. We will add this result in Table 4 and further analyze a range of $\beta$ values between 0 and 0.2 to present a more comprehensive ablation study.
>
> **Q7. It is recommended to further discuss the impact of non-iid data distribution.**
>
> Thank you for the insightful suggestion. As shown in Table 6, we already evaluated DictPFL under various degrees of data heterogeneity by adjusting the Dirichlet factor $\alpha$. While greater heterogeneity (lower $\alpha$) increases training time and communication overhead, DictPFL consistently maintains strong performance. This robustness is largely due to our HRC reactivation mechanism, which allows clients to share accumulated local gradients for reactivated parameters.
>
> To further demonstrate the effect, we compare “accumulative gradient sharing” versus “non-accumulative sharing,” measuring the resulting accuracy under similar training time. Our results (3-client ViT on Diabetic Retinopathy, $r=4$, $s=0.7$, $\tau=3$, $\beta=0.2$) show that omitting accumulated gradients notably reduces accuracy, particularly in more heterogeneous settings—because discarding small but meaningful gradients impairs learning for clients with diverse data. Overall, these results highlight that DictPFL can achieve robust performance across a wide range of data distributions.
>
> | $\alpha$  | Accumulative gradient sharing  | Non-accumulative sharing  |
> |-----------|--------------------------------|---------------------------|
> | 0.3       | 79.62%                         | 74.26%                    |
> | 0.6       | 80.28%                         | 76.13%                    |
> | 0.9       | 82.06%                         | 80.45%                    |

---

> > ### Comment · Reviewer_oqcz · 2025-08-05
> >
> > I sincerely thank the authors for the detailed response; it addressed my concerns, hence I raised my score.
> >
> > In the revised version, it's kindly recommended to use Figure 7 to support the claim (Lines 51–58), include results about how FedML-HE dynamically calculates sensitivity scores to encrypt, and the impact of non-iid data distribution.

---

### Note · Authors · 2025-08-13

We sincerely thank all reviewers for their **constructive feedback, positive ratings, and confirmation that their concerns have been fully addressed.** The reviewers recognized the importance and practicality of our proposed *DictPFL* for enabling efficient and privacy-preserving federated learning, highlighting:

- **Clear motivation and principled design** for improving performance without sacrificing privacy.
- **Significant efficiency gains**, increasing training time by less than a 2× factor compared to the plaintext counterpart, making privacy-preserving federated learning practical for real-world deployment.
- **Robustness against all gradient inversion attacks** by encrypting all transmitted gradients.

---
### Addressed Concerns:
- **Compare with FedML‑HE**: We will use experimental evidence (Fig. 7) to show FedML-HE's vulnerability when partial gradients are left unencrypted.
- **Additional attacks**: We clarified that *DictPFL* is robust to all existing attacks because all shared data is encrypted. We will highlight it further in our revision.

- **More baselines**: We confirmed that our chosen baselines remain standard in the HE-based FL research field. Recent work in 2024–2025 either built directly upon these widely adopted baselines or explored orthogonal optimizations, which are compatible with our *DictPFL*. We will discuss this more in our new revision.

-  **Algorithmic optimization vs. packing/quantization optimization**: We provided experiments showing that algorithmic optimization yields higher accuracy. These optimization directions are orthogonal, and *DictPFL* can further reduce costs when combined with packing methods.

-  **Non‑iid settings**: We evaluated across varying Dirichlet factors and demonstrated that the proposed HRC reactivation technology preserves important client-specific updates.
---
We are pleased that all reviewers indicated their concerns were resolved during rebuttal, with raising their scores or keeping positive scores. We commit to integrating all clarifications, additional experiments, and structural improvements (including a more concise introduction, clearer section organization, and standardized notations) into the final camera‑ready version.
We are grateful for the reviewers’, ACs’, and SACs’ time and insights. We believe our proposed *DictPFL*, by making full‑HE FL both secure and efficient, offers a practical path forward for deploying privacy-preserving FL in real‑world settings.

---

### Decision · Program_Chairs · 2025-09-17

**Decision:**

Accept (poster)

**Comment:**

This paper proposes **DictPFL**, a framework for efficient and private federated learning through selective encryption of model weights; with experiments across vision and NLP tasks showing clear improvements over prior art. The **final scores** of the paper read as 1 × Accept and 2 × Borderline Accept, an increase over the initial evaluations. After rebuttal and discussion, reviewers indicated that most concerns were addressed, and their overall stance is positive. The AC agrees with the reviewers’ evaluations, sees merit in the proposed algorithms, and therefore recommends **Accept**. Congratulations.

The AC encourages the authors to revise the paper based on the discussions during the rebuttal period. Finally, during the AC–reviewer discussion, it was noted that the **capacity of the model is inherently reduced by design**: the fixed dictionary and decomposition may constrain adaptation, particularly in heterogeneous settings. The authors are encouraged to acknowledge and discuss this limitation in the revision.